# Text mining of Reddit posts: Using latent Dirichlet allocation to identify common parenting issues

**Elizabeth M. Westrupp**[1,2,3]*, **Christopher J. Greenwood**[1,2,4], **Matthew Fuller-Tyszkiewicz**[1,2], **Tomer S. Berkowitz**[1,2], **Lauryn Hagg**[1], **George Youssef**[1,2,4]

**1** School of Psychology, Deakin University, Geelong, Victoria, Australia, **2** The Centre for Social and Early Emotional Development, Melbourne, Victoria, Australia, **3** Judith Lumley Centre, La Trobe University, Melbourne, Victoria, Australia, **4** Centre for Adolescent Health, Murdoch Children's Research Institute, Melbourne, Victoria, Australia

* elizabeth.westrupp@deakin.edu.au

**Data Availability Statement:** We draw on publicly available data from Reddit. Due to the nature of the data, we were not able to obtain individual participant consent. Instead, we applied for a

## Abstract

Parenting interventions offer an evidence-based method for the prevention and early intervention of child mental health problems, but to-date their population-level effectiveness has been limited by poor reach and engagement, particularly for fathers, working mothers, and disadvantaged families. Tailoring intervention content to parents' context offers the potential to enhance parent engagement and learning by increasing relevance of content to parents' daily experiences. However, this approach requires a detailed understanding of the common parenting situations and issues that parents face day-to-day, which is currently lacking. We sought to identify the most common parenting situations discussed by parents on parenting-specific forums of the free online discussion forum, Reddit. We aimed to understand perspectives from both mothers and fathers, and thus retrieved publicly available data from r/Daddit and r/Mommit. We used latent Dirichlet allocation to identify the 10 most common topics discussed in the Reddit posts, and completed a manual text analysis to summarize the parenting situations (defined as involving a parent and their child aged 0–18 years, and describing a potential/actual issue). We retrieved 340 (r/Daddit) and 578 (r/Mommit) original posts. A model with 31 latent Dirichlet allocation topics was best fitting, and 24 topics included posts that met our inclusion criteria for manual review. We identified 45 unique but broadly defined parenting situations. The majority of parenting situations were focused on basic childcare situations relating to eating, sleeping, routines, sickness, and toilet training; or related to how to respond to child negative emotions or difficult behavior. Most situations were discussed in relation to infant or toddler aged children, and there was high consistency in the themes raised in r/Daddit and r/Mommit. Our results offer potential to tailor parenting interventions in a meaningful way, creating opportunities to develop content and resources that are directly relevant to parents' lived experiences.

formal Exemption from Ethical Review from The Deakin University Human Research Ethics Committee (DUHREC). This process involved declaring that we would not share our data in order to protect any identifying features in the full dataset (e.g., full quotes of posts that might be directly searched online to reveal the original poster). Researchers who wish to obtain access to our dataset are able to make a request to the following: Human Research Ethics Office, Deakin University 221 Burwood Hwy Burwood, VIC 3125 AUSTRALIA Phone: +61 3 9251 7123 Email: research-ethics@deakin.edu.au Project: HEAG-H 140_2019: Internet text analysis of parenting situations.

**Funding:** The author(s) received no specific funding for this work.

**Competing interests:** The authors have declared that no competing interests exist.

## Introduction

A key limitation of community-based parenting interventions is that they are usually offered as one-size-fits-all packages, which stands in contrast to evidence that parents differ in the techniques that are most applicable to them [1–4], and therefore require tailored approaches [5]. Tailoring may take into account parents' needs or their day-to-day parenting context. To-date, tailoring in parenting interventions has focused on parents' perceived needs through adapting or adding intervention modules to be relevant to specific parent or child subpopulations [6–8], or determined by individual parent results from baseline surveys [9–11]. In contrast, tailoring related to parenting context has been minimal, despite the potential for this approach to increase parent engagement and intervention efficacy, in light of low rates of homework completion in parenting interventions [3], and recognized limitations in the human ability to generalize skills outside of specific learning contexts [12]. To enable a more systematic way of tailoring interventions to specific parenting situations, and thus facilitate parents to immediately implement intervention concepts in their daily lives, the field requires an accurate map of the wide range of day-to-day parenting situations that parents may need support with. The current study aims to meet this need, by investigating the most common issues that draw parents online to discuss their experiences of parenting with other parents.

Parenting interventions offer a strong evidence-based method for the prevention or early intervention of childhood mental health problems [9, 13]. Mental disorders rank as one of the highest causes of the global burden of disease [14]. To assist prevention efforts, the World Health Organization Nurturing Care Framework published in 2018 called for greater investment in adult education to enhance 'nurturing care' and thus promote positive early childhood development [13]. Despite many decades of investment and research, the long-term and population-level effectiveness of parenting programs has been lower than expected [1]. Children are influenced by their parents through multiple pathways, including modeling of behavior [15, 16], the emotional climate of the family [15], the nature and quality of the home environment [17], and via their parents' functioning and parenting practices [18–20]. Parenting interventions often target one or more of these domains, with the intention of modifying early determinants of children's socio-emotional development.

The field of parenting interventions is at an important juncture. Despite established efficacy for those who use the programs, parenting interventions have had very low reach (<10% population) [21, 22], particularly for fathers, working mothers, and disadvantaged families [3]. Most parenting interventions also struggle with low rates of parent engagement, adherence, homework completion, and participation [3, 23, 24]. A review of the qualitative evidence related to barriers for parent involvement in parenting programs found that parents identified two common reasons for dropping out: lack of fit with the therapist, and the nature of the program content or delivery method, for example, disliking the group activities, feeling that the content wasn't relevant, or that their needs were not recognized [25]. Fathers also identified mother-oriented services as a key barrier [25]. The next wave of parenting interventions need to address these barriers by designing more widely acceptable and flexible interventions, and through motivating and engaging parents by ensuring that content is flexibly tailored to a range of parent contexts and needs.

There are a number of different ways that community interventions can be tailored; for example, according to identified *need* or *context*. In regards to tailoring according to parent needs, there are two key approaches. The first approach has been to modify established interventions to meet the needs of specific parent sub-groups, such as parents of children with autism spectrum disorders [6], incarcerated parents [7], or parents struggling with sibling conflict [8]. This type of whole-of-intervention tailoring aims to increase the relevance of

intervention content to the perceived needs of the targeted sub-group of parents, but does not systematically adapt content to needs or the context of individual parents. A second approach has been tested more recently, and involves adapting the allocation of intervention content based on parent-level needs. This is illustrated in an online parenting intervention program where parents are provided with individualized recommendations regarding the specific sub-set or order of modules they should complete based on their results from a baseline parenting survey [9–11]. This approach has promising results [9–11].

However, to-date, there has been much more limited focus on tailoring according to parents' context. Although relatively neglected, there may be great potential to enhance parent engagement and learning through tailoring interventions to context, i.e., to specific parenting situations. In general, core intervention concepts within parenting interventions tend to be introduced and described at a higher-order conceptual level, or in relation to set hypothetical parenting examples. In face-to-face individual and group interventions, some degree of tailoring to context may occur in the sense that practitioners may discuss intervention concepts in relation to situations that parents raise during the session [26–28]. However, in most manualized parenting programs, there is only limited time to discuss parent-led examples, and this is even more limited in group-based or online programs. The fact that only a few examples are typically discussed, and that (in group settings) the examples may not even be relevant to individual parents, means that parents must later, and on their own, recall the higher-level concepts, and then reflect on how they might best be adapted to their own parenting situations. This mental gymnastics often must occur *in-the-moment* as parents attempt to change their usual patterns of responding to their child, while actually interacting with their child.

Imagine the possibility that parents can instead select a recent parenting situation that they have experienced in the past 24 to 48 hours and that they found difficult to manage. Instead of core concepts being described generically, they could be discussed directly in relation to the parent-selected example. In this case, it is likely that the parents' attention and interest would be heightened, given that the situation is immediately relevant to them. Description of content in relation to recent, known, examples is also likely to reduce parents' cognitive load, by reducing their working memory requirements; parents no longer need to consider and remember the parameters of a hypothetical example while simultaneously learning about new parenting concepts and techniques, and then attempting to problem-solve how they could adapt it to their own relevant example. Instead, if intervention content is tailored to a recent parenting situation, it is likely they will have opportunity to implement the tailored intervention concepts when the situation occurs again, given that many parenting situations are often repeated, particularly those that draw parents online to discuss or seek advice from other parents.

Tailoring interventions to specific parenting situations requires a detailed understanding of the common parenting situations and issues that parents face day-to-day. To our knowledge, this is currently lacking. The current study therefore aims to identify common parenting situations experienced by parents on a daily basis, which parents themselves identify as being difficult to manage. Our study seeks to better understand the types of parenting situations that motivate parents to reach out to other parents. Research shows that the internet has become a primary source of support for modern parents; parents go online, at any time of day or night, and seek parenting information and support from other parents on social media and online parenting forums [29].

Publically available data from online parenting forums offer an incredibly rich source of parent-centered data, expressed in parents' own words, and therefore uniquely untouched by researcher bias. Parent response options in quantitative surveys, and even in qualitative data collection procedures, are inherently influenced by researcher-design factors and choices. In comparison, online parenting forums reflect a naturally occurring situation where parents are

free to interact and discuss parenting issues without any constraints [30]. Therefore, data from online discussion forums offer considerable opportunity to create an accurate map of parenting situations to enable intervention tailoring.

The current study will use internet scraping to retrieve rich, publically-available data from the online discussion forum, Reddit. Reddit is one of the most popular social media platforms in the world, and the r/Daddit and r/Mommit forums have 163,000 and 106,000 members (at the time of writing), respectively. Importantly, the use of Reddit enables us to address the limitations of previously mother-centric research, by offering scope to investigate fathers' experiences in a rare father-specific forum where fathers are comfortable talking to other fathers about their parenting experiences [31]. We will utilise an increasingly popular machine learning text mining method, latent Dirichlet allocation (LDA) [32–35], to conduct a text analysis identifying the most common topics discussed in the Reddit posts. The LDA algorithm performs probability-based text mining to extract a set of topics from a corpus of text, based on patterns of words associated with each topic [32, 33]. LDA is a data-driven method, which minimizes researcher bias in the generation of topics, selection of key words, and ranking of relevant posts per topic [33]. This approach enables text analysis of very large datasets where more traditional qualitative text analysis approaches may not be feasible. LDA has been used to identify topics on Reddit discussion forums related to public health interest in same-sex marriage [36]; depression, mental health, treatment, and relationships [37]; Ebola, influenza, electronic cigarettes, and marijuana [38]; electronic cigarettes and Hookah use [39], and eating disorders [40]. LDA has also been used to investigate parenting, for example, one study compared LDA topics for mothers and fathers, and for fathers of preterm versus term children, derived from interview data assessing parents' reflective functioning [41].

The current study will use LDA and qualitative synthesis (i.e., manual text analysis) to address the following research question: What are the specific parenting situations being discussed within topics that emerge from text analysis of the Reddit r/Daddit and r/Mommit forums? Extraction of content from both of these parent-related forums will enable greater coverage of parenting issues for both fathers and mothers, in light of research showing gender differences in parenting topics discussed on Reddit [30]. It will also enable evaluation of potential differences in the concerns raised by each.

## Methods

### Ethics approval

The current study was approved by the Deakin University Human Ethics Advisory Group—Health (Project number: HEAG-H 140_2019). In line with HEAG-H advice, it was not possible to directly quote individual posts from publicly available datasets, thus data are presented in aggregated form only. Data were collected and used in accordance with Reddit's Terms and Conditions.

### Data analysis

Fig 1 outlines our processing and data analysis pipeline. Broadly, this comprised (a) data extraction, (b) data cleaning, (c) LDA topic modeling, and (d) qualitative synthesis. Of relevance, our approach to data cleaning and topic modeling is based on previously published work [42]. R software v3.6.1 [43] was used for all processing and analysis.

### Data extraction

We scraped publicly available data using 'RedditExtractoR' v2.1.5 [44] from two forums focused on experiences of mothers and fathers on Reddit (i.e., r/Daddit; r/Mommit). Data

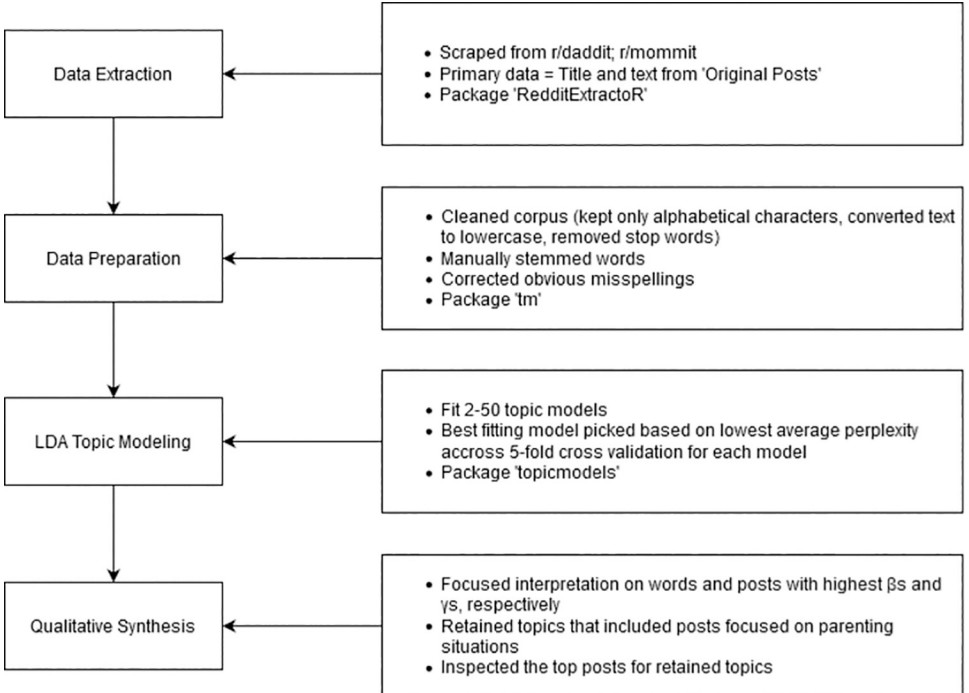

**Fig 1. Data processing and analysis pipeline.**

scraping was conducted on 11 February 2020 and we retained those posts that were the first of a particular thread (known as the 'original post') and that contained textual data in the body of the post, e.g., did not comprise just an image. Responses to the original posts were not included in analyses as this study aimed to only identify common parenting situations rather than describe the conversations and language used surrounding these situations. Due to limits in how data can be scraped from Reddit, only the first 10 pages of original posts from these forums could be scraped at any one time. Thus, we extracted 340 (r/Daddit) and 578 (r/Mommit) original posts that spanned an approximate 4–6 week period prior to our date of data scraping (r/Daddit earliest original post extraction date was 13 Jan 2020; r/Mommit earliest original post extraction date was 12 Dec 2019).

## Data preparation

The corpus of data were prepared for analysis using the 'tm' v0.7–6 [45] package. This process involved removing all non-alphabetical characters, punctuation, and blank spaces; converting all text to lowercase; removing all non-words; and removing "stop words" (e.g., "the", "is", "at", "which", "on"). We also manually 'stemmed' the corpus by reviewing the entire list of unique words and ensuring words with different suffixes (e.g., "happy", "happiness", "happiest") were coded as the same word (e.g., "happy"). During this review we also corrected obvious misspellings and international differences in spelling.

## LDA topic modelling

Common topics within the corpus were identified using latent Dirichlet allocation (LDA) [33], which is a common method used to identify themes in collections of scholarly textual data [32–35], as well as large sets of documents or texts in research or clinical contexts [46]. An

outline of the statistical methodology of LDA is presented in detail elsewhere [33, 47], but briefly described here. LDA is a Bayesian probabilistic modelling method that aims to identify the unknown number of latent topics that are assumed to underlie a body of text [33]. LDA draws from a Dirichlet distribution to generate distributions of probabilities that describe how (1) words (i.e., *word-topic-probabilities*) and (2) documents (i.e., *document-topic-probabilities*) are related to the latent topics underlying the dataset. Specifically, word-topic-probabilities are estimates of the probability a word is generated from a specific topic, whilst document-topic-probabilities are estimates of the probability that a topic has been generated in a specific document [48]. Inspection of the highest word-topic and document-topic probabilities for each topic can help characterize the theme of each latent topic. In the current study, we used LDA to find common topics within the corpus of original posts from r/Daddit and r/Mommit, and from these we aimed to identify parenting situations.

To conduct the LDA, we converted the corpus to a document-term-matrix, comprising rows representing each original post (i.e., document = original post) and columns representing each word in the corpus (i.e., term = word). Each cell in the document-term-matrix contains the frequency of times a specific word (defined by the column) occurred in a specific post (defined by the row). From this document-term-matrix, the entire corpus was represented, including patterns of words that commonly occur together within the same post.

The LDA was conducted using the 'topicmodels' v0.2–8 package [49]. As per Kosinski, Wang [50], LDA hyperparameters were set to delta = 0.1 and alpha = 50/k (where k is the number of LDA topics being estimated). To identify the number of LDA topics that best fit the patterns within the corpus, we first estimated a 2-topic model and then sequentially increased the number of LDA topics being modelled until we estimated a 50-topic model. We then selected the best fitting model based on the *perplexity* value for each model; a common method for evaluating model fit in LDA models [33, 51], where models with lower perplexity are considered to fit the data better than models with higher perplexity. Our estimate of perplexity for each model was derived as an average of the perplexity across a five-fold cross validation process. Within this process, the corpus of original posts was randomly split into five portions with a model generated (i.e., trained) on four of the portions, and then validated on the fifth portion. This process was repeated until each portion had been used for validation once. We also used the 'LDAvis' v0.3.2 package [52] to obtain the percentage of tokens (i.e. words) contributing to a specific topic.

## Qualitative analysis and interpretation

As mentioned previously, to interpret the topics derived from the optimal model, we relied on two metrics that form the basis of interpretation for LDA models. The first is a matrix of values that quantify the probability that each *word* on the corpus would be generated from a specific topic (i.e., $\beta$ matrix; higher $\beta$ = the word is more likely to occur in the topic). The second is a matrix of values that quantifies the proportion of words in an *original post* estimated to be generated by a specific topic (i.e., $\gamma$ matrix; higher $\gamma$ = the original post is more aligned with topic). Specifically, we focused interpretation on the 10 most relevant posts per topic according to $\gamma$ values. We completed a qualitative synthesis of the topics and original posts via a manual text analysis to interpret each topic and characterize specific parenting situations described in original posts within each topic. This procedure involved classifying each post according to whether they met a pre-specified definition of a 'parenting situation', defined as follows: (1) the post referenced a scenario involving a child aged 0–18 years; and (2) the scenario involved a parent and their child; and (3) referenced a potential or actual difficulty or issue, for which the parent was giving or seeking advice in how to manage or improve. Posts were also only

included if one or more of the 10 topic words was used in a way that conveyed meaning relevant to the central themes of a given post and topic, rather than being peripheral.

## Results

### LDA results

Fig 2 presents the perplexity values for each of the topic model scenarios, modeling 2–50 LDA topics, based on the number of topics retained in our LDA. Dots represent perplexity scores for each of the 5-fold cross-validation models for each topic. The blue line represents the average perplexity scores across the 5-fold cross validation. The lowest average perplexity was for a model with 31 topics (mean perplexity = 917.64). This 31-topic model was found to be the best performing LDA model as it had the lowest average perplexity of all the models (see S1 Table). The β values for the 10 most probable words associated with each of the 31 LDA topics, and the range of $\gamma$ values for the top 10 posts contributing to each of the 31 topics, are presented in S1 Fig and S2 Table, respectively. The average probability across all documents and the average and range of probabilities for the top 10 documents for each topic is presented in S3 Table. On inspection of the top 10 original posts related to each topic (i.e., with the highest $\gamma$ values), 24 of the 31 LDA topics contained posts that met the inclusion criteria in terms of being consistent with our pre-specified definition of a parenting situation, and meaningfully reflecting the topic words (i.e., with the highest β values).

### Parenting situations

Table 1 summarizes the LDA topic words, themes, and description of parenting situations for the 24 LDA topics that were included in the final analysis. The percentage of tokens contributing to each topic was relatively balanced, ranging from 2.6% (topic 31) to 4.1% (topic 2). Just eight of the 24 LDA topics included posts where parents were giving advice, either recounting their own recent experiences to illustrate a specific successful parenting moment that they

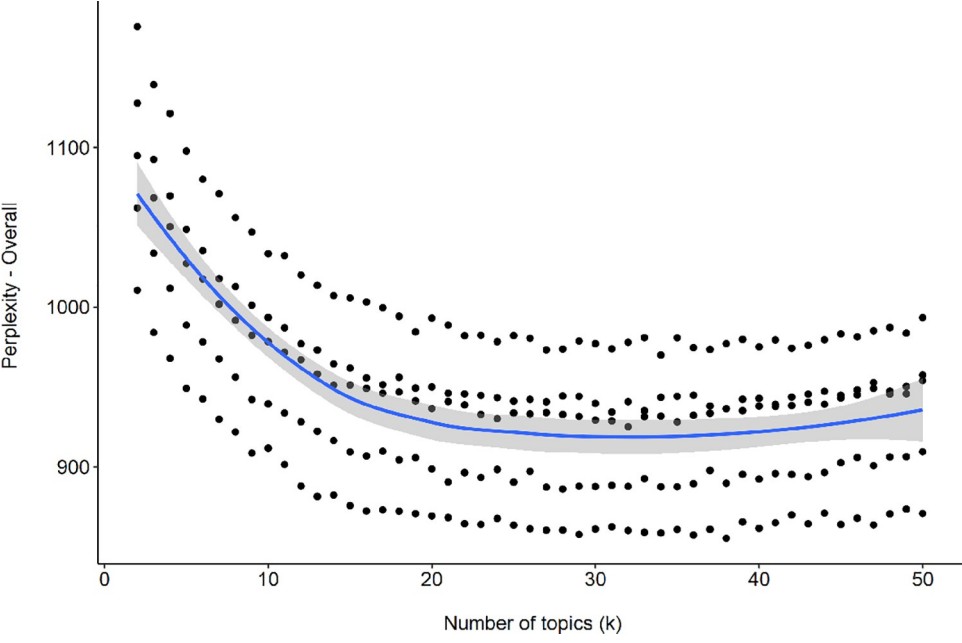

**Fig 2. Perplexity scores as a function of number of topics estimated.**

**Table 1. Summary of topic words, themes, and situation descriptions related to parenting situations described in Reddit parenting posts.**

| Topic (% of tokens) | LDA topic words | Topic themes | Descriptions of parenting situations |
|---|---|---|---|
| 2 (4.1%) | *Sleep, night, time, wake, bed, hour, put, nap, minute, crib* | Waking in night | Child sleep: waking in night wanting to play, feed or be held. |
| 3 (2.8%) | *Get, just, nappy, use, train, toddler, toilet, bad, run, poo* | Sickness and toilet training | Use of strategies for managing child with 'bad' cold or other sickness (e.g., runny nose). |
| | | | Issues or stages of toilet training. |
| 4 (2.8%) | *Day, bit, one, today, little, spend, start, bad, part, small* | Child refusing food or whining | Child refusing one food/part of a meal |
| | | | Child spending day crying/fussy; whinging if parent says no to something they want. |
| 5 (3.1%) | *Girl, boy, one, want, find, daughter, little, look, interest, anything* | Gender stereotypes | Parent concern about gender stereotypes (boy/girl): e.g., son teased for wanting to wear nail polish; parent wants child to be involved in sports, not just gender-stereotypical activities. |
| 7 (2.9%) | *Old, year, month, now, almost, sure, seem, still, last, half, yesterday* | Managing multiple children and sibling conflict. | Children (of different ages in months/years) only want to be picked up by mum. |
| | | | Child is acting younger than age (in months/years)—sleep or toilet training regression after birth of sibling. |
| | | | Managing multiple children of different ages with different needs and conflict. |
| 8 (3.2%) | *Will, partner, work, try, just, take, advice, anxiety, new, still* | Managing child emotions | Parent trying to manage child emotions/anxiety in new situations or teaching new skill (e.g., toilet training). |
| 9 (3.4%) | *Feel, like, just, much, morning, guilt, still, issue, better, though* | Parent guilt about time to self | Parent feeling guilty about wanting time to themselves, away from child. |
| 10 (3.7%) | *Say, tell, want, talk, know, school, let, made, break, ok* | How to talk to upset children | How to talk to kids when they're upset, want something they can't have, or won't do as asked. |
| 11 (3.3%) | *Feed, month, try, bottle, eat, breast, milk, give, breastfeed, formula* | Child refusing food | Child refusing to eat, breastfeed or feed from bottle. |
| 12 (3.5%) | *Friend, really, can, get, move, since, new, good, support, live* | Talking to children about change | Talking to child about moving to new daycare (e.g. leaving friends). |
| 13 (3.1%) | *Playroom, put, much, look, also, watch, come, dog, away, toy* | Setting rules | How to manage rules around safe rooms to play in the house. |
| | | | How to manage rules for putting toys away after play. |
| 15 (2.8%) | *Amp, even, know, use, see, xb, sister, phone, wonder, lie* | Child lying and phone use | Managing a child who has lied about something they did. |
| | | | Managing child's use of phone. |
| 17 (3.0%) | *Help, go, time, need, can, get, please, advice, appreciate, great* | Bed-time and juggling multiple children | Child refusing to go to bed on time; can't go to sleep. |
| | | | Advice on activities can do to manage multiple children at home. |
| 20 (3.1%) | *Get, hold, walk, cry, start, head, sit, hand, arm, push* | Physical support to child learning or upset | How to hold toddler when crying/having tantrum. |
| | | | How to support child learning to walk. |
| | | | Child hitting herself when upset and crying. |
| | | | Managing when a child prefers (to be held when crying) by one parent over another. |
| 21 (2.9%) | *Can, story, love, book, read, yo, hear, way, mean, oh* | Books to introduce difficult subjects | Child asking about death while reading books. |
| | | | Books to prepare a child for separation from parents. |
| 22 (3.0%) | *Baby, first, know, week, new, born, recommend, hair, newborn, able* | How to support a crying baby | How to support baby crying during tummy time. |
| 23 (3.7%) | *Like, son, think, never, ask, know, love, thing, start, always* | Managing child and parent autonomy; responding to child negative emotions and behaviour | How to respond to child always asking 'why' about everything. |
| | | | Thinking about how to manage strong feelings without always hitting their child. |
| | | | Should parents ask baby's permission before giving massage. |
| | | | Feeling judged for wanting time away from son. |
| | | | Child rude, disrespectful when asked to do something or when parent says no. |
| | | | Managing child lying about use of their phone . |

*(Continued)*

**Table 1.** (Continued)

| Topic (% of tokens) | LDA topic words | Topic themes | Descriptions of parenting situations |
|---|---|---|---|
| 24 (3.3%) | *Go, month, back, want, just, end, daughter, come, happen, hear, around* | Child develop-mental stages | Weaning a child from breastfeeding. |
| | | | Child going backwards in toileting. |
| | | | Child sleep and tantrums backwards (regression) after birth of sibling. |
| 25 (3.6%) | *Back, doctor, now, take, went, since, see, us, hospital, come* | Managing child sleep | Daughter will not sleep through the night; went to see doctor. |
| 26 (3.1%) | *Go, time, get, try, just, thing, frustrate, lot, easy, still* | Child or parent frustration | Baby frustrated trying tummy time. |
| | | | Father frustrated with daughter at bed-time or crying only with him. |
| | | | Parent frustrated child won't listen to "no". |
| | | | Son always climbing on something, parent worried about falls. |
| 28 (4.0%) | *Day, home, work, time, stay, leave, week, take, job, daycare* | Creating routine | Mum at home and trying to create routine for the day. |
| 29 (2.7%) | *Can, idea, anyone, us, good, buy, hard, lot, share, usual, favourite* | Strategies for calming child | Ideas for good strategies to calm child. |
| 30 (3.5%) | *Child, year, want, parent, family, don, house, us, come, live* | Child wants own way | Parent feels child wants their own way all the time. |
| 31 (2.6%) | *Tip, guy, advice, without, thank, car, look, infant, seat, appreciate* | Burping baby | Advice for how to best burp baby. |

hoped others might learn from (e.g., just found a good strategy for clearing their sick child's congestion), or some parents posted with more generic advice (i.e., not specific to a recent parenting experience, but based on their accumulated parenting experience), such as providing advice on strategies for specific situations (e.g., toilet training advice). The remainder of the topics involved posts where the parent was seeking advice from other parents on the Reddit parenting forums, most commonly in the format of a parent describing a specific parenting situation and asking others for advice on how they might handle it.

Based on the manual text analysis of posts within the LDA topics, we identified 45 unique but broadly defined parenting situations. The most prevalent themes from LDA topics meeting our inclusion criteria involved parents managing parenting situations related to child sleep, sickness, toilet training, child food refusal or fussiness, children whining when they didn't get what they wanted, gender stereotyping, managing multiple children/siblings, and managing children who are showing a regression in their behavior in relation to age. There were two very common higher-order themes, each identified in more than a third of the parenting situations, relating to (1) managing basic childcare situations, i.e., related to eating, sleeping, routines, sickness, toilet training; and (2) how to respond to child negative emotions and/or difficult behavior. Some topics were generated from parenting situations based on just one post e.g., Topic 31's sole post was seeking advice on burping a baby. Other topics were generated from parenting situations based on a number of posts e.g., Topic 2 had multiple posts on child sleeping habits.

## Parent and child characteristics

Table 2 summarizes the child ages and subreddit source for each of the included topic themes. The majority of the included posts (80%) referred to a child or children of infant or toddler age. Posts about children 4 years and older were focused on child use of technology, supporting children in gender non-stereotypical activities (e.g., sports, make-up), managing sibling conflict or the needs of multiple children at home, and using bibliotherapy to talk to children about change or potentially upsetting subjects. In relation to the source of posts, one third of the included posts were from the r/Daddit forum, while the remainder were from r/Mommit.

**Table 2. Child ages and subreddit source reflected in included topic themes.**

| Topic | Topic themes | Child ages[a] | subreddit source |
|---|---|---|---|
| 2 | Waking in night | Infant (8); Toddler (2) | r/Mommit; r/Daddit |
| 3 | Sickness and toilet training | Infant (4); Toddler (5) | r/Mommit; r/Daddit |
| 4 | Child refusing food or whining | Infant (1); Toddler (2); Young child (1) | r/Mommit; r/Daddit |
| 5 | Gender stereotypes | Young child (6) | r/Mommit; r/Daddit |
| 7 | Managing multiple children and sibling conflict. | Infant (2); Toddler (4); Young child (2) | r/Daddit |
| 8 | Managing child emotions | Toddler (2) | r/Mommit |
| 9 | Parent guilt about time to self | Infant (2); Toddler (1) | r/Mommit |
| 10 | How to talk to upset children | Toddler (1); Young child (3) | r/Mommit; r/Daddit |
| 11 | Child refusing food | Infant (6); Toddler (1) | r/Mommit; r/Daddit |
| 12 | Talking to children about change | Toddler (1) | r/Mommit |
| 13 | Setting rules | Toddler (2); Young child (1) | r/Mommit; r/Daddit |
| 15 | Child lying and phone use | Older child (3) | r/Daddit |
| 17 | Bed-time and juggling multiple children | Toddler (1); Young child (3) | r/Mommit |
| 20 | Physical support to child learning or upset | Infant (2); Toddler (2) | r/Mommit; r/Daddit |
| 21 | Books to introduce difficult subjects | Toddler (1); Young child (1) | r/Mommit |
| 22 | How to support a crying baby | Infant (1) | r/Mommit |
| 23 | Managing child and parent autonomy; responding to child negative emotions and behaviour | Infant (1); Toddler (2); Young child (2) | r/Mommit; r/Daddit |
| 24 | Child developmental stages | Infant (1); Toddler (2); Young child (1) | r/Mommit; r/Daddit |
| 25 | Managing child sleep | Toddler (1) | r/Mommit |
| 26 | Child or parent frustration | Infant (1); Toddler (4); | r/Mommit; r/Daddit |
| 28 | Creating routine | Infant (1) | r/Mommit |
| 29 | Strategies for calming child | Infant (1); Toddler (1) | r/Mommit |
| 30 | Child wants own way | Infant (1); Toddler (1) | r/Daddit |
| 31 | Burping baby | Infant (1) | r/Daddit |

[a]Child age categories were coded as follows: Infant (0–11 months); Toddler (1–3 years); Young child (4–10 years); Older child (10+ years). Numbers in parentheses indicate the number of children mentioned within posts in the given topic related to that age group.

The majority of the LDA topics included posts from both sources, meaning that most of the identified parenting situations were discussed by mothers and fathers online. The two most common higher-order themes (i.e., managing basic childcare situations and responding to child negative emotions and/or difficult behavior) were also reflected in both r/Mommit and r/Daddit posts. The identity (e.g., gender) of the person writing each post was not known, and is therefore not able to be described.

Whilst we focused our interpretation on topics that met our inclusion criteria, interested readers can find a summary of themes for posts not meeting the inclusion criteria for the current study in S4 Table. In summary, parents posted on parenting subreddits for a large number of reasons, including medical issues not related to being a parent; managing relationships with family members other than their children, such as the relationship with their partner; and managing a range of situations related to being a parent, but not specific to a parent-child interaction, including moving house, seeking advice around their own feelings and wellbeing, navigating special occasions and holidays, their child's school, and managing issues related to pets and children.

## Discussion

### Principal results

The objective of the current study was to support the necessary first step in developing parenting interventions that are tailored to parents' context, by providing a detailed understanding

of the common parenting situations and issues that parents face day-to-day. In this study, we sought to identify the most common day-to-day parenting situations discussed online on the r/Daddit and r/Mommit subreddit forums to inform meaningful tailoring in parenting interventions. Using the innovative and increasingly popular machine learning approach, latent Dirichlet allocation (LDA) [33], we were able to exploit the incredibly rich source of person-centered parenting data available online, in order to identify parenting situations that are most applicable to parents day-to-day. The LDA topic modeling extracted 31 LDA topics, of which 24 met our inclusion criteria in terms of being related to a difficult parenting situation involving their child. From these, we identified 45 unique but broadly defined parenting situations involving a parenting issue or difficult interaction between a parent and their child aged 0–18 years.

Our study is the first we are aware of to systematically describe the most topical and common parenting situations that parents seek advice and support around. Over two-thirds of the LDA topics meeting our inclusion criteria for analysis related to two primary themes. First–managing basic childcare situations, i.e., related to eating, sleeping, routines, sickness, toilet training; and second–related to advice on how to respond to child negative emotions and/or difficult behavior. Although there is little in the way of published evidence documenting the specific parenting examples used in parenting interventions, it is the experience of the authors that these examples tend to relate to situations involving the second theme, i.e., related to managing child negative emotions and difficult behavior. Our results suggest that many parents are also very interested in how to manage children in day-to-day childcare tasks. The parent-child relationship is equally formed around management of these daily rhythms, as it is in the management of difficult child behavior or emotions. Thus, it seems appropriate that future parenting interventions could include content focused on parent-child interactions across the breadth of parenting situations described in our results.

The inclusion of posts from r/Daddit and r/Mommit meant that our results reflected perspectives from both mothers and fathers. Despite clear evidence that fathers are central in influencing child developmental outcomes [53, 54], including child mental health outcomes [19, 55], fathers are chronically underrepresented in parenting research and parenting interventions (<20% of attendees) [56]. A systematic review of fathers' representation in observational parenting studies identified that just 10% of 667 parenting studies included results separately for fathers; 1% were focused solely on fathers compared to 36% focused solely on mothers [57]. Our study findings address this gap by describing topics discussed by mothers and fathers. Although we identified some differences in LDA topics discussed in each of the forums, on the whole, there was a high level of consistency in the themes raised in r/Daddit and r/Mommit. However, our findings suggest that parents raise different questions depending on the age of their child, suggesting that parenting interventions should tailor intervention content to be age-specific. Further, the vast majority of included posts (80%) related to infant or toddler aged children. These findings suggest that early childhood may be an important time for intervention when parents are particularly open to receiving support and advice.

We note that just 24 of the 31 LDA topics identified met our inclusion criteria for analysis as a parenting situation. The remaining posts traversed a wide range of themes, including managing family situations and relationships in the extended family context, parents' own health concerns, and household-related issues. Parents increasingly use online tools in a multitude of ways to assist with parenting, including connecting through social media, assisting with managing family life, and providing multiple sources of online information [29]. Our data reflect that parents use Reddit to access support from other parents for a wide range of reasons. Although some of the LDA topics identified in the current study were not deemed to

be immediately relevant in terms of tailoring parenting interventions, they may be useful for understanding how parents access support online.

## Applications for tailoring parenting interventions

E-interventions, delivered online or via smartphone apps, offer potential for overcoming cost, time, and geographic barriers faced by traditional interventions [58]. However, results to-date have been disappointing. A systematic review showed that current technology-based parenting interventions are still under-servicing key groups, such as fathers, non-urban parents, and parents from a low socio-economic background [59]. For the most part, technology-based parenting interventions have not changed the format or content from traditional face-to-face, module-based approaches [58]. Tailoring in technology-based parenting interventions has been minimal to-date, despite evidence that that engagement is strongest when interventions are flexibly tailored to be maximally relevant to individual parents [58, 60]. Findings from the current study provide scope to address these limitations. We identify 45 of the most commonly discussed parenting situations on the parenting-specific forums of Reddit. Our results provide a detailed picture of the common scenarios that mothers and fathers face day-to-day, that could be applied to tailor resources in population-based parenting interventions.

## Strengths and limitations

The use of LDA topic modeling and manual text analysis was both a strength and potential limitation of the current study. LDA offers a powerful data-driven method for analyzing large datasets, and may discern patterns in qualitative data that becomes infeasible to analyze with traditional qualitative coding approaches as data quantity increases. However, LDA lacks specific recommendations around best practice approaches in particular settings. For example, there is no single method for determining the most appropriate number of LDA topics to model within the data [32]. Further, decisions made in the data preparation steps to clean the data prior to running the LDA modeling have potential to introduce researcher bias and impact the results.

Our findings may also be limited by the scope offered in the Reddit parenting forums. For example, while widely used, not all parents use Reddit. Further, for Reddit users, some parents may be more likely to comment on others' posts, rather than create an original post themselves. In both cases, the experiences of these parents will not be reflected in our results. There may also be bias in the subjects that parents raise for discussion on Reddit, for example, based on parents' prior interpretation of what is acceptable within particular forums. In addition, some topics may be less likely to be identified by the LDA process, such as those related to sensitive or less common situations, or where a range of different words are used to refer to the same meaning, thus making it less likely to be identified as a topic by the LDA modelling. Further research efforts to augment publicly available posts with data directly obtained from Reddit users about their demographics and other parenting-related constructs may provide added context with which to appraise the potential generalizability of results obtained from online forum conversations.

Our results reflect data collected from the Reddit platform in 2020, but are likely to be generalizable to other online forums used by parents, such as Facebook and Instagram. There is emerging evidence investigating the way in which parents use other social media sites, including Facebook parenting groups [61, 62]. However, Reddit is uniquely amenable to topic modelling, given that all posts are publicly available, which is not the case with Facebook and Instagram. We expect that the way in which parents discuss parenting issues would be relatively consistent across platforms, but there is currently no evidence to assess this claim. This

could be tested by future research investigating whether there are systematic differences between parent users of different platforms, such as demographic characteristics (e.g., parent user age, gender, country of residence, spoken language).

## Conclusions

Our results support the use of LDA text mining for the purpose of understanding broad themes discussed in online parenting forums. We identified 45 unique parenting situations describing a wide range of parenting contexts, but most commonly related to basic childcare situations, i.e., related to eating, sleeping, routines, sickness, toilet training; or related to advice about how to respond to child negative emotions and/or difficult behavior. The current study was novel in analyzing parents' own words to understand the most common day-to-day parenting issues experienced by parents. Internet forums represent a rich source of potentially unbiased data for researchers, offering a method for observing and analyzing naturally-occurring adult conversations online. These findings offer potential to tailor parenting interventions in a meaningful way, creating opportunities to develop content and resources that are directly relevant to parents' lived experiences.

## Supporting information

**S1 Table. Mean perplexity values across 5-fold cross-validations for K-topic solutions.**
(DOCX)

**S2 Table. Mean, standard deviation, and range of gamma values for the top 10 posts contributing to each topic.**
(DOCX)

**S3 Table. Range and average document-topic-probabilities for the top ten documents for each topic.**
(DOCX)

**S4 Table. Summary of themes from posts not meeting the study inclusion criteria.**
(DOCX)

**S1 Fig. β values for the 10 most probable words associated with each of the 31 LDA topics.**
(TIF)

## Author Contributions

**Conceptualization:** Elizabeth M. Westrupp, Lauryn Hagg, George Youssef.

**Data curation:** Elizabeth M. Westrupp, Christopher J. Greenwood, Matthew Fuller-Tyszkiewicz, Tomer S. Berkowitz, Lauryn Hagg, George Youssef.

**Formal analysis:** Elizabeth M. Westrupp, Christopher J. Greenwood, Matthew Fuller-Tyszkiewicz, Lauryn Hagg, George Youssef.

**Investigation:** Elizabeth M. Westrupp, Matthew Fuller-Tyszkiewicz, Tomer S. Berkowitz, Lauryn Hagg, George Youssef.

**Methodology:** Elizabeth M. Westrupp, Christopher J. Greenwood, Matthew Fuller-Tyszkiewicz, Lauryn Hagg, George Youssef.

**Project administration:** Elizabeth M. Westrupp, Tomer S. Berkowitz.

**Software:** Christopher J. Greenwood, Matthew Fuller-Tyszkiewicz, Lauryn Hagg, George Youssef.

**Supervision:** Elizabeth M. Westrupp, George Youssef.

**Validation:** Christopher J. Greenwood, Matthew Fuller-Tyszkiewicz, Lauryn Hagg, George Youssef.

**Writing – original draft:** Elizabeth M. Westrupp, Christopher J. Greenwood, Matthew Fuller-Tyszkiewicz, Lauryn Hagg, George Youssef.

**Writing – review & editing:** Elizabeth M. Westrupp, Christopher J. Greenwood, Matthew Fuller-Tyszkiewicz, Tomer S. Berkowitz, Lauryn Hagg, George Youssef.

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
