## [Decision Letter · Decision Letter 0]

27 Apr 2021

PONE-D-21-07563

Text mining of Reddit posts: Using latent Dirichlet allocation to identify common parenting issues

PLOS ONE

Dear Dr. Westrupp,

Thank you for submitting your manuscript to PLOS ONE. After careful consideration, we feel that it has merit but does not fully meet PLOS ONE’s publication criteria as it currently stands. Therefore, we invite you to submit a revised version of the manuscript that addresses the points raised during the review process.

We look forward to receiving your revised manuscript.

Kind regards,

Ali B. Mahmoud, Ph.D.

Academic Editor

PLOS ONE

Journal Requirements:

2) PLOS ONE has specific requirements for studies using personal data from third-party sources, including social media, blogs, other internet sources, and phone companies (https://journals.plos.org/plosone/s/submission-guidelines#loc-personal-data-from-third-party-sources). These requirements include confirming data are collected and used in accordance with the company or website’s Terms and Conditions, obtaining appropriate ethics or data protection body review, and ensuring appropriate consent from individuals whose data are used in research. In this case, please ensure that your Ethics statement is in compliance with guidelines, and that you have complied with the company's (i.e., Reddit) Terms and Conditions, with appropriate permissions.

3) Please upload a copy of Figure 2, to which you refer in your text. If the figure is no longer to be included as part of the submission please remove all reference to it within the text.

4) We note that you have indicated that data from this study are available upon request. PLOS only allows data to be available upon request if there are legal or ethical restrictions on sharing data publicly. For more information on unacceptable data access restrictions, please see http://journals.plos.org/plosone/s/data-availability#loc-unacceptable-data-access-restrictions.

Reviewers' comments:

Reviewer's Responses to Questions

**Comments to the Author**

1. Is the manuscript technically sound, and do the data support the conclusions?

Reviewer #1: Partly

Reviewer #2: Yes

2. Has the statistical analysis been performed appropriately and rigorously? 

Reviewer #1: Yes

Reviewer #2: Yes

3. Have the authors made all data underlying the findings in their manuscript fully available?

Reviewer #1: No

Reviewer #2: No

4. Is the manuscript presented in an intelligible fashion and written in standard English?

Reviewer #1: Yes

Reviewer #2: Yes

5. Review Comments to the Author

Reviewer #1: The article topic is exciting and essential for all families and young people who want to form a family. However, there are some notes.

1. The research lacks a clear explanation of the methodology, while the Methods section focuses more on the LDA model's steps.

2. The size of the data used is relatively small, and it is preferable to increase the number or cite similar articles that use the same or less data volume.

3. The authors have deleted the stop words, but in table one, I find some words that are considered stop words like will, can, since and others. So is there a reason not to delete them, and if there is a reason, the authors should write it.

4. The authors stemmed the words manually; why?

Note: some tools do it automatically

5. Authors do not write the percentage of each topic to the overall size of the dataset. Through this percentage, we can find out the actual weight of each topic.

Note: the authors referred to this point on page 9, row 164, but did not implement it

6. The authors wrote on page 13, row 256, "Specifically, we focused interpretation on the 10 most relevant posts per topic according to values.". It is better to specify a percentage rather than a fixed number.

7. page 15, row 285 to 291, more explanation needs to be mentioned

8. Table 1 column "LDA topic words ", some words", not refer to the topic themes like

topic 4. day, one, today

topic 7. last, yesterday

topic 9. like, much

topic 10. ok

it is general words that can be on different topics. Personally prefer to eliminate it as stop words

, and some words are not clear like yo and oh on topic 21,

9. The authors wrote on page 24, "there is no single method for determining the most appropriate number of LDA topics to model within the data." this is true, for that we sometimes resort to the help of experts in the field of study to determine the topics' numbers "human judgment ". I recommend that authors use this method to determine the number of topics because the 31 topics selected in this study are a big number, and some of them refer to the same themes.

Reviewer #2: The authors present a data mining-based approach to identify common parenting situations discussed by parents on parenting platforms such as r/Daddit and r/Mommit. LDA is used to identify the most common topics in a date set extracted and scraped from relevant websites.

The work is technically sound and the findings are interesting and has potential to feed future applications/chatbots that may benefit and reach to larger number of parents in different geographical locations and variety of languages.

I do have some concerns that requires to be answered before the paper is accepted for publication:

1- Why were only 2-topics being estimated at the beginning of the study? Isn't that more topics would contribute to an informative results?

2- How are the number of models being increased till the number reached 50 topics? The method used must be shared.

3- What is the perplexity value of each model not being shared?

4- On what basis the parenting situations were defined? Any logic behind the identification of those situations? Probably previous experiences or data from key informants has fed in such selection. For instance, posts that involves a child aged 0-18 may not involve parenting issues. How it was decided to included as a parenting situation?

5-Figure 2 that represents the perplexity values for topic models scenarios is missing from the article.

6-How did your study conclude that parents use Reddit to access support from other parents? I haven't seen any data that supports such a claim.

7-Some of LDA topics, that were considered not immediately relevant to parenting interventions, may have used different keywords to inquire about important parenting matters. How to ensure such topics are included in the analysis?

8-The authors should acknowledge the fact that Daddit and Mommit may have been used by specific communities/ countries so the issue of generalisation of results must be discussed in depth.

6. PLOS authors have the option to publish the peer review history of their article (what does this mean?). If published, this will include your full peer review and any attached files.

Reviewer #1: No

Reviewer #2: No

---

## [Author Response · Author response to Decision Letter 0]

19 Jul 2021

Please see attached formatted document with table showing reviewer comments and our response. The section pasted here might be harder to read.

The article topic is exciting and essential for all families and young people who want to form a family. However, there are some notes. 

1. The research lacks a clear explanation of the methodology, while the Methods section focuses more on the LDA model's steps. We have now added some more technical details about the LDA methodology in the method section (see “LDA topic modelling”) and also provided references for technical descriptions. The edited section is presented below (on page 12): 

“An outline of the statistical methodology of LDA is presented in detail elsewhere [47, 48], but briefly described here. LDA is a Bayesian probabilistic modelling method that aims to identify the unknown number of latent topics that are assumed to underlie a body of text.[33] LDA draws from a Dirichlet distribution to generate distributions of probabilities that describe how (1) words (i.e., word-topic-probabilities) and (2) documents (i.e., document-topic-probabilities) are related to the latent topics underlying the dataset. Specifically, word-topic-probabilities are estimates of the probability a word is generated from a specific topic, whilst document-topic-probabilities are estimates of the probability that a topic has been generated in a specific document [49]. Inspection of the highest word-topic and document-topic probabilities for each topic can help characterize the theme of each latent topic.”

2. The size of the data used is relatively small, and it is preferable to increase the number or cite similar articles that use the same or less data volume.

 The issue of sample size for LDA remains unclear in the literature, with no firm direction about required N nor sufficient empirical appraisal of impacts of sample size on obtained results. Indeed, a key challenge is that sample size calculations for machine learning algorithms cannot be calculated a priori because model performance is entirely based on the strength of the signals underlying a dataset, which can only be identified using machine learning itself. This means that the dataset sizes are not comparable since these will be modeling different sets of topics. We also note that many studies do not report on the number of documents they have used. In fact, we have just completed a scoping review of 47 papers using LDA in the psychological sciences (under review), and found only 34% of the papers reported this information. Some examples of papers not reporting:

Ruiz, N., Witting, A., Ahnert, L., & Piskernik, B. (2020). Reflective functioning in fathers with young children born preterm and at term. Attachment & Human Development, 22(1), 32-45. https://doi.org/10.1080/14616734.2019.1589059

Barry, A. E., Valdez, D., Padon, A. A., & Russell, A. M. (2018). Alcohol Advertising on Twitter—A Topic Model. American Journal of Health Education, 49(4), 256-263. https://doi.org/10.1080/19325037.2018.1473180

Carpenter, J., Crutchley, P., Zilca, R. D., Schwartz, H. A., Smith, L. K., Cobb, A. M., & Parks, A. C. (2016). Seeing the 'big' picture: Big data methods for exploring relationships between usage, language, and outcome in Internet intervention data. Journal of Medical Internet Research, 18(8), e241-e241. https://doi.org/10.2196/jmir.5725

Hemmatian, B., Sloman, S. J., Cohen Priva, U., & Sloman, S. A. (2019). Think of the consequences: A decade of discourse about same-sex marriage. Behavior Research Methods, 51(4), 1565-1585. https://doi.org/10.3758/s13428-019-01215-3

Gerber, M. S. (2014). Predicting crime using Twitter and kernel density estimation. Decision Support Systems, 61, 115-125. https://doi.org/10.1016/j.dss.2014.02.003

As such, we argue that more relevant is the topics that are generated and the validity of those topics in understanding or being used for their intended purpose. 

We also wish to emphasize that the results from our study reflect the level of data available via Reddit for parenting conversations, and are not a subsample of possible content. This both constrains our capacity to expand the analysis, but also provides validity to results as they are reflective of the breadth and content of current parenting conversations on Reddit. 

3. The authors have deleted the stop words, but in table one, I find some words that are considered stop words like will, can, since and others. So is there a reason not to delete them, and if there is a reason, the authors should write it.

 Please note that we have used the stop words function in the tm package, which uses a list of stop words that can be found here: http://snowball.tartarus.org/algorithms/english/stop.txt. The authors of this package explain that some words are not included as stop words since they are common homonyms (e.g., “can”, “will”). Nevertheless, we do not believe that removing these potential words would alter the interpretations of these topics, which should remain stable even with stop words removed given these words should not impact interpretation of the theme of a topic (i.e., because these words are not meaningful). Given the complexity of the data that is presented on an online forum, we appreciate some level of noise may be generated from these words. However, despite this noise, we believe that the results are still meaningful, with minimal impact on our aim to identify the parenting situations. 

4. The authors stemmed the words manually; why? Note: some tools do it automatically We appreciate the availability of some algorithms that can be used to automatically stem words in a corpus. However, we are mindful that automatic stemming approaches can reduce precision because this can lead to dissimilar meaning words being treated as the same stemmed word in analysis (see pp34 ref, as below, for examples). Given the nature of our dataset, and the potential for some overlapping words, we believed it was worth engaging in a manual stemming approach rather than automation of this task. For example, in our manual stemming approach there was clear advantage in discriminating some words that would automatically be stemmed to the same stem for the purpose of analysis (e.g., “grandfather”, “grandmother”, “granny”, “granddad”, “grandpa” = all stemmed to “grandparent”; whilst “granddaughter”, “grandchild”, “grandson” = all stemmed to “grandchild”). As such, we believe our manual approach was appropriate in this instance.

Reference: Christopher D. Manning, Prabhakar Raghavan, H. S. (2008). Introduction to Information Retrieval.

 

5. Authors do not write the percentage of each topic to the overall size of the dataset. Through this percentage, we can find out the actual weight of each topic.

Note: the authors referred to this point on page 9, row 164, but did not implement it.

 We have now provided the average probability and range of probabilities for the top 10 documents for each topic in Supplementary Table 3. Whilst this does provide an indication of the relationship between documents to topics, we note that the aim of this study specifically was to identify the situations that were commonly experienced by parent, rather than focusing on the latent topics that were estimated. Consequently, we believe that this information is best presented as supplementary material. For convenience, we have presented this table below (at the end of the document).

6. The authors wrote on page 13, row 256, "Specifically, we focused interpretation on the 10 most relevant posts per topic according to 𝛾 values.". It is better to specify a percentage rather than a fixed number.

 As noted in response 5, we have now provided a summary of the range and average document-topic-probabilities for the top 10 documents for each topic. However, as noted in our previous response we believe this information is best suited to the supplementary material given that the primary focus of the manuscript was to identify the parenting situations within each topic.

7. page 15, row 285 to 291, more explanation needs to be mentioned We have added additional detail to better summarize the key information presented in Table 1 (page 16):

“Just eight of the 24 LDA topics included posts where parents were giving advice, either recounting their own recent experiences to illustrate a specific successful parenting moment that they hoped others might learn from (e.g., just found a good strategy for clearing their sick child’s congestion), or some parents posted with more generic advice (i.e., not specific to a recent parenting experience, but based on their accumulated parenting experience), such as providing advice on strategies for specific situations (e.g., toilet training advice). The remainder of the posts involved the parent seeking advice from other parents on the Reddit parenting forums, most commonly in the format of a parent describing a specific parenting situation and asking others for advice on how they might handle it.”

 

8. Table 1 column "LDA topic words ", some words", not refer to the topic themes like

topic 4. day, one, today 

topic 7. last, yesterday

topic 9. like, much

topic 10. ok

it is general words that can be on different topics. Personally prefer to eliminate it as stop words, and some words are not clear like yo and oh on topic 21. We acknowledge that some of these words appear to be unrelated to the topics. However, we accept some level of “noise” within the dataset. Our revisions to the data were limited to stemming, so that patterns that emerged were likely to be a better reflection of how words relate, even if some words were unexpected. This was preferable – in our view – to overly interfering with the data to force a solution that made sense to us a priori. Moreover, given our study was aimed primarily on identifying the parenting situations that are commonly experienced, we believe that these examples will not impact on the identification of such situations. As such we accept that whilst not perfect, these represent common issues in collecting data from websites such as reddit and do not impact on our primary focus on identifying parenting situations.

9. The authors wrote on page 24, "there is no single method for determining the most appropriate number of LDA topics to model within the data." this is true, for that we sometimes resort to the help of experts in the field of study to determine the topics' numbers "human judgment ". I recommend that authors use this method to determine the number of topics because the 31 topics selected in this study are a big number, and some of them refer to the same themes. We thank the reviewer for this suggestion. Given that the primary aim of this study was to identify the parenting situations that were discussed within each topic, we believe that our data driven approach (which suggested a 31 topic model) was appropriate. In this case, having a large number of topics was not necessarily burdensome given that it was the identification of the situations that was the aim of the study, and that it was therefore possible that there could be more (or less) than 31 situations that could be identified within the topics. As noted in the discussion (page 23), we:

“...extracted 31 LDA topics, of which 24 met our inclusion criteria in terms of being related to a difficult parenting situation involving their child. From these, we identified 45 unique but broadly-defined parenting situations involving a parenting issue or difficult interaction between a parent and their child aged 0-18 years.”

As such, our approach was not reliant on identifying a small number of topics, and our inclusion criteria focused only on topics that comprised and there was clear differentiation within the situations. Consequently we believe our data driven approach appropriately addressed our research questions.

---

## [Decision Letter · Decision Letter 1]

7 Sep 2021

PONE-D-21-07563R1Text mining of Reddit posts: Using latent Dirichlet allocation to identify common parenting issuesPLOS ONE

Dear Dr. Westrupp,

Thank you for submitting your manuscript to PLOS ONE. After careful consideration, we feel that it has merit but does not fully meet PLOS ONE’s publication criteria as it currently stands. Therefore, we invite you to submit a revised version of the manuscript that addresses the points raised during the review process. Moreover, the current manuscript omitted responses and revision work concerning the comments made by Reviewer 2. Therefore, I invite you to address/respond to the comments by both reviewers cautiously in your next revision.

We look forward to receiving your revised manuscript.

Kind regards,

Ali B. Mahmoud, Ph.D.

Academic Editor

PLOS ONE

Journal Requirements:

Additional Editor Comments (if provided):

Reviewers' comments:

Reviewer's Responses to Questions

**Comments to the Author**

1. If the authors have adequately addressed your comments raised in a previous round of review and you feel that this manuscript is now acceptable for publication, you may indicate that here to bypass the “Comments to the Author” section, enter your conflict of interest statement in the “Confidential to Editor” section, and submit your "Accept" recommendation.

Reviewer #1: (No Response)

Reviewer #2: (No Response)

2. Is the manuscript technically sound, and do the data support the conclusions?

Reviewer #1: Partly

Reviewer #2: (No Response)

3. Has the statistical analysis been performed appropriately and rigorously? 

Reviewer #1: No

Reviewer #2: (No Response)

4. Have the authors made all data underlying the findings in their manuscript fully available?

Reviewer #1: No

Reviewer #2: (No Response)

5. Is the manuscript presented in an intelligible fashion and written in standard English?

Reviewer #1: Yes

Reviewer #2: (No Response)

6. Review Comments to the Author

Reviewer #1: 1. in my previous review, I wrote, "The research lacks a clear explanation of the methodology, while the Methods section focuses more on the LDA model's steps." I guess that the authors do not understand what I mean, so I will rewrite my review. Data extraction, data preparation, LDA topic modeling, and qualitative synthesis are general steps we apply in any LDA analysis. For this reason, the methodology paragraph should explain how the LDA modeling will be used to achieve the research goal. Otherwise, this research is nothing more than an application of LDA modeling and does not provide anything new.

2. There is not a complete stop words list; therefore, the possibility to update this list is available.

stopwords = nltk.corpus.stopwords.words('english')

stopwords.append('newWord')

or

stopwords = nltk.corpus.stopwords.words('english')

newStopWords = ['stopWord1','stopWord2']

stopwords.extend(newStopWords)

3. in my previous review, I wrote, "Authors do not write the percentage of each topic to the overall size of the dataset. Through this percentage, we can find out the actual weight of each topic." in this comment, I mean the topic not the document.

The authors can use tools like this "" ext-link-type="uri" xlink:type="simple">https://pyldavis.readthedocs.io/en/latest/modules/API.html" to get the percentage of each topic.

This percentage is significant to know which topics are more representative.

4. The appearance of stop words in the table of topics is unacceptable; thus, it must be deleted before applying the LDA modeling, as I mentioned earlier. Furthermore, the analysis is based on the appearance of the words within the same document and in different documents; consequently, when we delete any word, a new word will appear in the topic's list of words, which can change the meaning of this topic.

5. The authors have mentioned that "we note that the aim of this study specifically was to identify the situations that were commonly experienced by parent."

The identity of the situations will be identified through the results of LDA modeling; consequently, any problems in the application of the model will prevent us from reaching the real results "situations".

Reviewer #2: The author hasn't' responded to my review that was sent with the decision letter. Please ensure that the below points are addressed clearly the response letter.

The authors present a data mining-based approach to identify common parenting situations discussed by parents on parenting platforms such as r/Daddit and r/Mommit. LDA is used to identify the most common topics in a date set extracted and scraped from relevant websites.

The work is technically sound and the findings are interesting and has potential to feed future applications/chatbots that may benefit and reach to larger number of parents in different geographical locations and variety of languages.

I do have some concerns that requires to be answered before the paper is accepted for publication:

1- Why were only 2-topics being estimated at the beginning of the study? Isn't that more topics would contribute to an informative results?

2- How are the number of models being increased till the number reached 50 topics? The method used must be shared.

3- What is the perplexity value of each model not being shared?

4- On what basis the parenting situations were defined? Any logic behind the identification of those situations? Probably previous experiences or data from key informants has fed in such selection. For instance, posts that involves a child aged 0-18 may not involve parenting issues. How it was decided to included as a parenting situation?

5-Figure 2 that represents the perplexity values for topic models scenarios is missing from the article.

6-How did your study conclude that parents use Reddit to access support from other parents? I haven't seen any data that supports such a claim.

7-Some of LDA topics, that were considered not immediately relevant to parenting interventions, may have used different keywords to inquire about important parenting matters. How to ensure such topics are included in the analysis?

8-The authors should acknowledge the fact that Daddit and Mommit may have been used by specific communities/ countries so the issue of generalisation of results must be discussed in depth.

7. PLOS authors have the option to publish the peer review history of their article (what does this mean?). If published, this will include your full peer review and any attached files.

Reviewer #1: No

Reviewer #2: **Yes: **Eiad Yafi

---

## [Author Response · Author response to Decision Letter 1]

21 Oct 2021

Ali B. Mahmoud, Ph.D.

Wednesday, 15 September 2021

Dear Dr Mahmoud

Thank you for your support of our paper. We want to pass on our sincere apologies to Reviewer 2 for unintentionally omitting their comments in our submitted response document. We have now addressed all comments, and provide a detailed response to both Reviewer 1 and 2’s comments in the table below.

Warm regards

Dr Elizabeth Westrupp

Deakin University

---

## [Decision Letter · Decision Letter 2]

29 Nov 2021

PONE-D-21-07563R2Text mining of Reddit posts: Using latent Dirichlet allocation to identify common parenting issuesPLOS ONE

Dear Dr. Westrupp,

Thank you for submitting your manuscript to PLOS ONE. After careful consideration, the reviewers have recommended publication but also suggested a few minor corrections before the paper is fully accepted. Therefore, I invite you to submit a revised version of the manuscript that addresses the points raised during the review process.

If applicable, we recommend that you deposit your laboratory protocols in protocols.io to enhance the reproducibility of your results. Protocols.io assigns your protocol its own identifier (DOI) so that it can be cited independently in the future. For instructions see: https://journals.plos.org/plosone/s/submission-guidelines#loc-laboratory-protocols. Additionally, PLOS ONE offers an option for publishing peer-reviewed Lab Protocol articles, which describe protocols hosted on protocols.io. Read more information on sharing protocols at https://plos.org/protocols?utm_medium=editorial-emailutm_source=authorlettersutm_campaign=protocols.

We look forward to receiving your revised manuscript.

Kind regards,

Ali B. Mahmoud, Ph.D.

Academic Editor

PLOS ONE

Journal Requirements:

Reviewers' comments:

Reviewer's Responses to Questions

**Comments to the Author**

1. If the authors have adequately addressed your comments raised in a previous round of review and you feel that this manuscript is now acceptable for publication, you may indicate that here to bypass the “Comments to the Author” section, enter your conflict of interest statement in the “Confidential to Editor” section, and submit your "Accept" recommendation.

Reviewer #1: All comments have been addressed

2. Is the manuscript technically sound, and do the data support the conclusions?

Reviewer #1: Partly

3. Has the statistical analysis been performed appropriately and rigorously? 

Reviewer #1: Yes

4. Have the authors made all data underlying the findings in their manuscript fully available?

Reviewer #1: No

5. Is the manuscript presented in an intelligible fashion and written in standard English?

Reviewer #1: Yes

6. Review Comments to the Author

Reviewer #1: i still have the same comment

The appearance of stop words in the table of topics is

unacceptable; thus, it must be deleted before applying the

LDA modeling, as I mentioned earlier. Furthermore, the

analysis is based on the appearance of the words within the

same document and in different documents; consequently,

when we delete any word, a new word will appear in the

topic's list of words, which can change the meaning of this

topic.

the stop word forbids giving an explanation of the meaning of the topics

7. PLOS authors have the option to publish the peer review history of their article (what does this mean?). If published, this will include your full peer review and any attached files.

Reviewer #1: No

---

## [Author Response · Author response to Decision Letter 2]

21 Dec 2021

Dear Dr Mahmoud

Thank you for your support of our paper. We have explained below that we disagree with Reviewer 1’s perspective in regards to stop words. We have included a detailed explanation below, and note that our approach is backed up in a number of published papers. 

Warm regards

Dr Elizabeth Westrupp

Deakin University 

Reviewer #1: i still have the same comment

The appearance of stop words in the table of topics is

unacceptable; thus, it must be deleted before applying the

LDA modeling, as I mentioned earlier. Furthermore, the

analysis is based on the appearance of the words within the

same document and in different documents; consequently,

when we delete any word, a new word will appear in the

topic's list of words, which can change the meaning of this

topic. 

the stop word forbids giving an explanation of the meaning of the topics

OUR RESPONSE

We believe we have conducted an appropriate level of stopword removal. Our approach focussed on removing the most commonly recognised stopwords, as well as stopwords common to the online setting. We used a standard stopword list implemented in the ‘tm’ package for our study (https://rdrr.io/rforge/tm/man/stopwords.html) containing common stopwords. In addition, we added to this list our own list of stopwords relevant to the online setting (e.g. “goodbye”, “howdy”, “html”, “com”, “login”), and removed single letters and common combinations of two letter words that would appear after removal of punctuation (e.g., “nt”). 

Schofield et al (2017) conducted simulation studies examining the impact of stopwords on interpretation and found little benefit in a continuously iterative process of updating a stoplist and then re-estimating the model. They argue that this approach does not impact the interpretation of the topics, which usually rely in inspection of words which have the highest word-topic probabilities. They suggest that simply ignoring those stopwords, should they appear in the list (which they term “post-hoc stopword removal”), is entirely appropriate and does not invalidate the estimates mode. Specifically, they note in their conclusion (p435-436): 

“Generating a corpus-specific stoplist to remove rarer contentless words provides relatively little utility to training a model. To obtain the benefit of a stoplist, it suffices to remove the most frequent, obvious stopwords from a corpus without developing a specific stoplist for the problem setting. If these methods are not sufficient, we find that post-hoc stopword removal can significantly improve coherence while avoiding many of the efficiency and epistemological bias issues of iterative stoplist curation.”

Given that we have removed the most common stopwords as defined by well-used text mining packages, as well as words that would are specific to the online setting, we believe our topics are robust to the inclusion of any additional ad hoc stopwords identified post hoc. Moreover, we believe our findings are robust since our focus was on the parenting contexts identified within the topics. Consequently, we were not limited to interpreting any of the top words for each topic but rather interrogated the posts to identify the parenting situations . 

Reference:

Schofield, A., Magnusson, M., Mimno, D. (2017). Pulling out the stops: Rethinking stopword removal for topic models. 15th Conference of the European Chapter of the Association for Computational Linguistics, EACL 2017 - Proceedings of Conference, 2, 432–436. https://doi.org/10.18653/v1/e17-2069.

---

## [Decision Letter · Decision Letter 3]

29 Dec 2021

Text mining of Reddit posts: Using latent Dirichlet allocation to identify common parenting issues

PONE-D-21-07563R3

Dear Dr. Westrupp,

We’re pleased to inform you that your manuscript has been judged scientifically suitable for publication and will be formally accepted for publication once it meets all outstanding technical requirements.

Kind regards,

Ali B. Mahmoud, Ph.D.

Academic Editor

PLOS ONE

Additional Editor Comments (optional):

Reviewers' comments:

Reviewer's Responses to Questions

**Comments to the Author**

1. If the authors have adequately addressed your comments raised in a previous round of review and you feel that this manuscript is now acceptable for publication, you may indicate that here to bypass the “Comments to the Author” section, enter your conflict of interest statement in the “Confidential to Editor” section, and submit your "Accept" recommendation.

Reviewer #1: All comments have been addressed

2. Is the manuscript technically sound, and do the data support the conclusions?

Reviewer #1: Partly

3. Has the statistical analysis been performed appropriately and rigorously? 

Reviewer #1: Yes

4. Have the authors made all data underlying the findings in their manuscript fully available?

Reviewer #1: Yes

5. Is the manuscript presented in an intelligible fashion and written in standard English?

Reviewer #1: Yes

6. Review Comments to the Author

Reviewer #1: (No Response)

7. PLOS authors have the option to publish the peer review history of their article (what does this mean?). If published, this will include your full peer review and any attached files.

Reviewer #1: No

---

## [Editor Report · Acceptance letter]

6 Jan 2022

PONE-D-21-07563R3 

Text mining of Reddit posts: Using latent Dirichlet allocation to identify common parenting issues 

Dear Dr. Westrupp:

I'm pleased to inform you that your manuscript has been deemed suitable for publication in PLOS ONE. Congratulations! Your manuscript is now with our production department. 

Kind regards, 

on behalf of

Dr. Ali B. Mahmoud 

Academic Editor

PLOS ONE